# The Role of Nintedanib in the Treatment of Progressive Pulmonary Fibrosis of Autoimmune-Related Interstitial Lung Disease

Aulia Rahman Ardan  and Fariz Nurwidya *

Department of Pulmonology and Respiratory Medicine, Universitas Indonesia Faculty of Medicine—Persahahabatan Hospital, Jakarta 13230, Indonesia; aulia.rahman71@ui.ac.id
* Correspondence: fariz.nurwidya@ui.ac.id

**Abstract:** Interstitial lung disease (ILD), which is characterized by pulmonary fibrosis, is a diverse group of disorders. Nintedanib, an antifibrotic drug, is known to attenuate disease progression in ILD with progressive fibrosis, but its efficacy in autoimmune-disease-related ILD remains uncertain. We conducted a comprehensive search for relevant randomized controlled trials, systematic reviews and meta-analyses included in PubMed, ScienceDirect and Scopus databases as of 23 June 2022 and manually reviewed reference lists. Among the 689 titles and abstracts screened, 24 studies were considered, with 4 randomized controlled trials included in our review. Nintedanib, administered at 150 mg twice daily for 52 weeks, consistently slowed forced vital capacity decline. Enhanced efficacy was observed when combining nintedanib with immunomodulators, and the most common adverse effect was diarrhea. In conclusion, our study suggests that nintedanib is a safe option for mitigating the progression of autoimmune-disease-related ILD, providing valuable insights into its potential therapeutic role in this context.

**Keywords:** nintedanib; antifibrotic; interstitial lung disease; autoimmune disease

## 1. Introduction

Interstitial lung disease (ILD) is characterized by scarring (fibrosis) of the lungs and is part of a heterogenous group of disorders differentiated by different parameters, including histopathology, radiology and clinical findings [1]. Common manifestations of ILD are impaired gas exchange and restrictive lung defects [2]. Idiopathic pulmonary fibrosis (IPF) is the most common type of ILD and is characterized by progressive fibrosis in the lung parenchyma. Idiopathic pulmonary fibrosis is often associated with poor outcomes [1,2]. Other forms of ILD include those related to SLE, such as rheumatoid arthritis (RA) and systemic sclerosis (SSc). These autoimmune diseases have been known to cause connective-tissue-disease-related ILD (CTD-ILDs) [3].

The mechanism of CTD-ILD is sequences of inflammation in the lungs including the bronchioles, parenchyma and alveolus. The tissues in the lungs are rich in proteins and profibrotic elements promoting a cascade of activation and accumulation of connective tissue. When these inflammatory processes are not inhibited, they may lead to progressive fibrosing interstitial lung disease (PF-ILD), in which the disease becomes irreversible. Nevertheless, there is currently no universal definition of PF-ILD. However, recent studies defined progressive pulmonary fibrosis as meeting at least two of the three following criteria within 12 months: worsening respiratory symptoms; lung function deterioration; and an increase in fibrotic features in HRCT [4]. Correct identification and prompt treatment are important to have better outcomes [1,5].

Recent studies have investigated different approaches to reducing the progressivity of ILD [5–7]. The first approach to treating ILD is investigating the underlying cause and managing it appropriately, such as suppressing systemic inflammation found in autoimmune diseases or ensuring avoidance of irritants to the lungs. Furthermore, immunomodulatory drugs including corticosteroids with or without supplementary immunosuppressant

therapy have been proven to reduce inflammatory activity within the lungs [1,5,6]. A study by Johannson et al. explains that in patients with acute hypersensitivity pneumonitis, the use of prednisone showed improvement and a slower rate of decline in forced vital capacity (FVC) [5]. Therapy of ILD aims to slow down the disease progression and improve pulmonary outcomes [5–7].

The use of nintedanib, an antifibrotic drug, is known to reduce disease progression specifically in ILD with progressive fibrosis [5]. Nintedanib is an oral intracellular tyrosine kinase inhibitor that inhibits the signaling cascade involved in fibrogenesis. Nintedanib use for PF-ILD has been correlated with a reduced rate of decline in FVC, maintaining quality of life and better prognosis outcomes [7–9]. Nintedanib's ability to inhibit Lck moderates the release of profibrotic and immune-stimulating mediators. This modulation effect is predicted to be important in the treatment of CTD-ILDs [5]. The purpose of this study was to identify whether nintedanib is effective in reducing the progressivity of autoimmune-disease-mediated ILD.

## 2. Method

A mini-review was conducted using various databases. Searches were performed in PubMed, ScienceDirect and Scopus using the terms (nintedanib) AND ((ILD) OR (interstitial lung disease)) AND ((progressive) OR (progressivity)), as shown in Table 1. The number of search results obtained for each database can be seen in Table 2. Studies that were duplicates were deleted, and studies were selected based on eligibility criteria, title and abstract screening according to PICO and full-text selection. All selected studies were subjected to a critical appraisal using appraisal tools from the Oxford Center for Evidence-Based Medicine (OCEBM) for intervention studies.

**Table 1.** PICO framework.

| Patient/Problem (P) | Intervention (I) | Comparison (C) | Outcome (O) |
|---|---|---|---|
| Patients with autoimmune disease and proven ILD | Nintedanib | Placebo/none | ILD progression |
| Type of clinical question | Therapy | | |
| Study design | RCT, systematic review and/or meta-analysis | | |

**Table 2.** Literature search strategy.

| Database | Search Strategy | Results |
|---|---|---|
| PubMed | (nintedanib) AND ((ILD) OR (interstitial lung disease)) AND ((progressive) OR (progressivity)) | 42 |
| ScienceDirect | (nintedanib) AND ((ILD) OR (interstitial lung disease)) AND ((progressive) OR (progressivity)) | 492 |
| Scopus | (nintedanib) AND ((ILD) OR (interstitial lung disease)) AND ((progressive) OR (progressivity)) | 200 |

All studies found were selected using inclusion and exclusion criteria. The inclusion criteria in question were as follows: (a) randomized controlled trial, systematic review or meta-analysis (b) published in 2017–2022 and (c) in accordance with the determined PICO. Meanwhile, the exclusion criteria used were as follows: (a) research written in a language other than English or Indonesian, (b) not available as a full text and (c) research results not related to the determined PICO.

Article searches were carried out on 3 databases, and 734 articles were found. After the duplication check, there were 689 articles remaining. The articles were then selected through title and abstract screening until there were 26 articles remaining. The article selection was continued with full-text selection, and 2 articles were found not to be available as a full text.

Subsequently, 20 studies were excluded due to inappropriate study design, outcome and subjects. Additionally, 1 article was excluded because it was in the form of a letter to the editor. The selection process is presented in Figure 1.

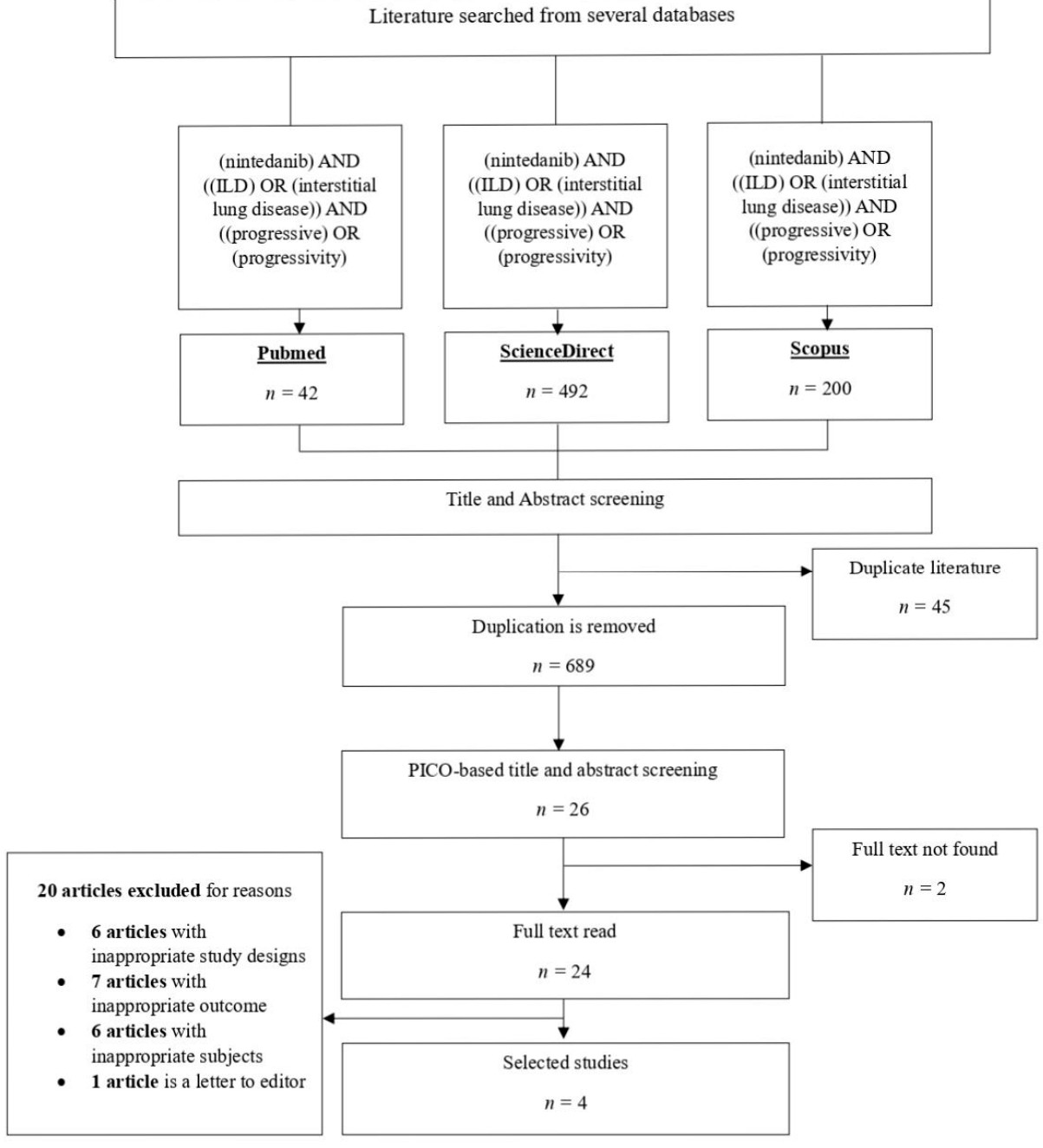

**Figure 1.** Search strategy flowcharts.

### 3. Result

The selected articles were articles with a study design in the form of a randomized controlled trial. The populations under study were adult autoimmune-related interstitial lung disease patients. The ILD patients referred to are those with >10% pulmonary fibrosis on high-resolution computed tomography (HRCT) and those who had forced vital capacity percent predicted (FVC%). Adult patients were patients aged 18 years or older. Patients were given an intervention of nintedanib 150 mg twice daily for at least 52 weeks. Selected studies were expected to demonstrate the effect of nintedanib on ILD disease progression.

The characteristics of each study can be seen in Table 3. A critical appraisal of the methods used in the selected articles was conducted based on the Oxford Center of Evidence-Based Medicine (CEBM) guidelines for randomized controlled trials.

There were four studies that went through the CEBM critical appraisal for randomized controlled trials (Tables 4 and 5). The majority of studies met the five existing criteria. However, there was one study that did not explain how many patients were lost to follow-up and one study that had a loss-to-follow-up rate of more than 20%.

Matteson E.L. et al. (2022) described the rate of decline in FVC over 52 weeks to be lower in the nintedanib group than the placebo group (difference 102.7 mL/year [95% CI 23.2, 182.2]; nominal $p$ = 0.012) [10]. Acute exacerbations of ILD or death occurred less frequently in the nintedanib group than the placebo group (HR 0.58 [95% CI 0.27 1.27]; nominal $p$ = 0.17). No heterogeneity effect was detected between the nintedanib group and placebo group across subgroups based on ILD diagnosis ($p$ = 0.91). The most frequent adverse event was diarrhea, which was reported at a higher rate in the nintedanib group than the placebo group. Nausea, vomiting, constipation, decreased appetite, weight decrease and abdominal pain were also more frequently reported in the nintedanib group [10].

Highland K.B. et al. (2021) found that the adjusted mean annual rate of decline in FVC was lower in patients taking mycophenolate at baseline with nintedanib than in those taking a placebo, with a difference of 26.3 mL per year [95% CI −26.9 to 80.6] [11]. Meanwhile, in patients not taking mycophenolate at baseline, the adjusted mean annual rate of decline in FVC was also lower than in those taking a placebo, with a difference of 55.4 mL per year [95% CI 2.3 to 108.5]. They also state that the most common adverse event was diarrhea, which was more frequently found in the nintedanib group than the placebo group [11].

Kuwana M. et al. (2020) found that in Japanese patients, the adjusted annual rate of FVC decline over 52 weeks was lower in the nintedanib group than the placebo group, with a difference of 4.67 mL/year (95% CI −103.28, 112.63) [12]. They also state that there was no effect on heterogeneity between Japanese and non-Japanese patients ($p$ = 0.49). The most common adverse events of nintedanib were mild-to-moderate liver and gastrointestinal disorder [12].

Azuma A. et al. (2021) found that the rate of FVC decline over 52 weeks was consistent between Asian and non-Asian patients in the placebo group [13]. The effect of nintedanib on reducing the rate of FVC decline over 52 weeks was also consistent between Asian (difference: 44.3 mL/year [95% CI: −32.8, 121.4]) and non-Asian patients (difference: 39.0 mL/year [95% CI: −5.1, 83.1]) (treatment-by-time-by-subgroup interaction, $p$ = 0.91). The most frequent adverse event, diarrhea, was reported in a similar proportion of non-Asian and Asian patients in the placebo group and the nintedanib group [13].

**Table 3.** Study characteristics.

| Author (Year) | Study Design | Population | Intervention | Outcome | Results |
|---|---|---|---|---|---|
| Matteson E.L., Kelly C., Distler J.H.W., Seibold J.R., Mittoo S. et al., (2022) | Randomized controlled trial | Patients with a fibrosing autoimmune-disease-related interstitial lung disease (ILD) | I: patients who received nintedanib 150 mg twice a day<br><br>C: patients who received placebo | Rate of decline in FVC and adverse events over 52 weeks in the subgroup with autoimmune-disease-related ILDs | • The rate of decline in FVC over 52 weeks in the nintedanib group was lower than that in the placebo group.<br>• The number of patients who had acute exacerbations of ILD, progression of ILD or died in the nintedanib group was lower than that in the placebo group.<br>• No heterogeneity was detected in the effect of nintedanib versus the placebo.<br>• The most frequent adverse event reported in nintedanib and placebo groups was diarrhea. |
| Highland K.B., Distler O., Kuwana M., Allanore Y., Assassi S et al., (2021) | Randomized controlled trial | Patients aged 18 years or older with systemic-sclerosis-associated interstitial lung disease (SSc-ILD) and onset of first non-Raynaud's symptom less than 7 years prior to study | I: patients who received 150 mg of oral nintedanib twice daily<br><br>C: patients who received placebo | The rate of decline in FVC over 52 weeks according to mycophenolate use at baseline | • Nintedanib reduced the progression of interstitial lung disease in patients with SSc-ILD who were and were not using mycophenolate at baseline.<br>• Adverse events were reported in subgroups by mycophenolate use at baseline.<br>• The combination of mycophenolate and nintedanib offers a safe treatment for SSc-ILD. |
| Kuwana M., Ogura T., Makino S., Homma S., Kondoh Y. et al., (2020) | Randomized controlled trial | Patients from Japan aged 20 years or older and patients from other countries aged 18 years or older who had a diagnosis of SSc, onset of the first non-Raynaud's symptom within 7 years prior to study, >10% lung fibrosis within 12 months FVC > 40% and diffusion capacity of lung for carbon monoxide | I: patients who received oral nintedanib 150 mg twice daily<br><br>C: patients who received placebo | Efficacy and safety of nintedanib in Japanese patients with systemic-sclerosis-associated interstitial lung disease (SSc-ILD) | • Nintedanib slowed the progression of ILD in both Japanese and non-Japanese patients.<br>• There was no heterogeneity detected between Japanese and non-Japanese patients with SSc-ILD. |
| Azuma A., Chung L., Behera D., Chung M., Kondoh Y. et al., (2021) | Randomized controlled trial | SSc patient with onset of first non-Raynaud's symptom less than 7 years prior to study and extent of fibrotic ILD > 10% | I: patients who received nintedanib 150 mg twice daily<br><br>C: patients who received placebo | Efficacy and safety of nintedanib in patients of Asian race | • Nintedanib had a consistent benefit in slowing the progression of SSc-ILD as nintedanib had a consistent benefit in reducing the rate of FVC decline in Asian and non-Asian patients.<br>• The adverse event profile was similar in both Asian and non-Asian patients.<br>• The adverse events were manageable for most patients. |

**Table 4.** Validity of the included studies.

| Questions | Matteson E.L., Kelly C., Distler J.H.W., Seibold J.R., Mittoo S. et al., (2022) | Highland K.B., Distler O., Kuwana M., Allanore Y., Assassi S. et al., (2021) | Kuwana M., Ogura T., Makino S., Homma S., Kondoh Y. et al., (2020) | Azuma A., Chung L., Behera D., Chung M., Kondoh Y. et al., (2021) |
|---|---|---|---|---|
| *Was the assignment of patient to treatments randomised?* | Yes | Yes | Yes | Yes |
| *Were the groups similar at the start of the trial?* | Yes | Yes | Yes | Yes |
| *Aside from the allocated treatment, were groups treated equally?* | Yes | Yes | Yes | Yes |
| *Were all patients who entered the trial accounted for? And were they analysed in the group which they were randomised?* | Unclear | No | Yes | Yes |
| *Were measures objective or were the patients and clinicians kept "blind" to which treatment was being received?* | Yes | Yes | Yes | Yes |

**Table 5.** *Applicability* overview.

| Questions | Matteson E.L., Kelly C., Distler J.H.W., Seibold J.R., Mittoo S. et al., (2022) | Highland K.B., Distler O., Kuwana M., Allanore Y., Assassi S. et al., (2021) | Kuwana M., Ogura T., Makino S., Homma S., Kondoh Y. et al., (2020) | Azuma A., Chung L., Behera D., Chung M., Kondoh Y. et al., (2021) |
|---|---|---|---|---|
| *Is my patient so different to those in the study that the results cannot apply?* | No | No | No | No |
| *Is the treatment feasible in my setting?* | Yes | Yes | Yes | Yes |
| *Will the potential benefits of treatment outweigh the potential harms of treatment for my patient?* | Yes | Yes | Yes | Yes |

## 4. Discussion

We will first discuss the effect of nintedanib on pulmonary function. The studies included support the beneficial effect of nintedanib on pulmonary function [10–13]. Pulmonary function is measured through pulmonary function tests (PFTs), which provide insight into the condition of the lung structure and function. A spirometry test quantifies lung function through the results of forced expiratory volume (FEV), forced vital capacity and FEV/FVC ratio. Restrictive abnormalities in the lungs are observed in ILD, and patients may present with a reduction in FVC of less than 80% with a ratio of FEV/FVC > 70% [14,15]. The rate of decline in FVC was used as a parameter in predicting the progressivity of ILD across the studies [10–13].

The effect of nintedanib on the rate of decline in FVC was measured from the start of treatment until over 52 weeks. The studies had a similar drug regimen, wherein nintedanib was consumed 150 mg twice daily [10–13]. The same dosing of nintedanib was used for IPF and other forms of ILD [16]. Matteson E.L. et al. found that across the autoimmune-related ILDs, nintedanib consumption shows a significantly slower rate of decline over 52 weeks (−75.9 mL/year) compared to the placebo (−178.7 mL/year) [10]. Particularly in patients with rheumatoid-associated ILD (RA-ILD), nintedanib use showed a significant difference in the FVC rate of decline (difference 117.9 mL/year) in comparison to other autoimmune-disease-related ILDs. However, caution should be taken in interpreting these results due to the sample size of each group [10]. Findings by Kuwana M. et al. show similar results, where the use of nintedanib over 52 weeks in patients with SSc-ILD slowed the

rate of decline in FVC by −52.4 mL/year compared to −93.3 mL/year in placebo-treated patients [12]. Furthermore, according to Azuma A. et al., in Asian and non-Asian patients, nintedanib use was able to reduce the rate of FVC by −55.6 mL/year and −51.6 mL/year, respectively [13]. In comparison, both placebo-treated Asian and non-Asian populations had a higher FVC rate of decline at −99.9 mL/year and −90.6 mL/year, respectively [13]. In the studies mentioned above, nintedanib was shown to be effective, slowing down the progression of FVC decline at more than twice the rate seen in patients not given nintedanib. Previous studies found that nintedanib's efficacy can be observed not only in autoimmune-related ILDs but also in other forms of ILD, such as IPF [16,17].

The interaction between nintedanib and other drugs has been observed in several studies. In Matteson E.L. et al.'s study, it was found that there was no heterogeneity between the use of DMARDs and/or glucocorticoids with nintedanib [10]. Interestingly, in the same study, the FVC decline was significantly greater in patients taking a combination of nintedanib and DMARDs and/or glucocorticoids [10]. Additionally, according to Highland et al., nintedanib in tandem with mycophenolate led to a much slower rate of FVC decline (−40.2 mL/year) compared to that in patients that only consumed nintedanib (−63.9 mL/year) [11]. Mycophenolate and glucocorticoids are immunomodulators, which suggests that in addition to nintedanib slowing down the rate of fibrosis, the additional benefits of the inflammatory process being suppressed help in slowing down ILD progression [18,19]. The increased efficacy seen with the combination of nintedanib and immunomodulatory drugs shows that there is a positive interaction that benefits the patients.

Overall, there are minimal side effects associated with nintedanib consumption. Across the studies that were included, the most commonly observed adverse effects were diarrhea and nausea [10–13]. Furthermore, based on Matteson E.L. et al.'s study, there were more adverse events found in sample groups taking a combination of nintedanib and other drugs [10]. Similarly, in a previous study, common effects of nintedanib consumption were diarrhea and nausea [20]. There have been occurrences of an increase in liver enzymes; however, this increase does not meet Hy's law and does not pose a risk of fatal drug-induced liver injury [10,20]. From these studies, we can observe that nintedanib use is relatively tolerable and the beneficial effect of nintedanib consumption outweighs the adverse effects.

## 5. Conclusions

Nintedanib exerts its effect in slowing down the progression of autoimmune-related interstitial lung disease. Daily consumption of nintedanib over 52 weeks not only shows a slower rate of FVC decline but has further compounding effects when taken with immunomodulatory drugs. Furthermore, nintedanib is relatively safe, with diarrhea as the most frequent adverse effect. At the time of diagnosis, prompt treatment with nintedanib can delay progression of fibrosis in autoimmune-related interstitial lung disease.

There is no doubt about the benefits of nintedanib consumption for ILD progression. However, limited studies have explored its specific effects on different types of autoimmune diseases, such as SLE. The authors recommend that clinical trials be carried out in Indonesia to investigate the use of nintedanib in daily practice and targeted towards different types of autoimmune-disease-related ILDs.

**Author Contributions:** Conceptualization, A.R.A. and F.N.; methodology, A.R.A. and F.N.; software, A.R.A.; validation, A.R.A. and F.N.; formal analysis, A.R.A. and F.N.; investigation, A.R.A. and F.N.; resources, A.R.A. and F.N.; data curation, A.R.A. and F.N.; writing—original draft preparation, A.R.A.; writing—review and editing, A.R.A. and F.N.; supervision, F.N.; project administration, A.R.A. All authors have read and agreed to the published version of the manuscript.

**Funding:** This research received no external funding.

**Conflicts of Interest:** The authors declare no conflict of interest.

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
