# Peer review of "The Role of Nintedanib in the Treatment of Progressive Pulmonary Fibrosis of Autoimmune-Related Interstitial Lung Disease"

_2673-527X, doi:10.3390/jor3040019_

Round 1

Reviewer 1 Report

Comments and Suggestions for Authors

In this review, the authers discussed the efficacy of using Nintedanib for the treatment of autoimmune-related ILD progressive pulmonary fibrosis; the manuscript is well written but:

 I think the authors did not show any novelty in their review, there is already published review and meta-analysis study that addresses the effect of nintedanib in progressive pulmonary fibrosis “Nintedanib in Progressive Pulmonary Fibrosis A Systematic Review and Meta-Analysis”

Author Response

Manuscript titled “THE ROLE OF NINTEDANIB IN REDUCING PROGRESSION OF AUTOIMMUNE DISEASE RELATED INTERSTITIAL LUNG DISEASE”

We would like to express our deep gratitude to the Editor and Reviewers for the important feedbacks and suggestions. We have addressed the Reviewers comments point by point below. We provide clean and tracked changes version of the revised manuscript. Our responses are as following.

Reviewer 1

In this review, the authers discussed the efficacy of using Nintedanib for the treatment of autoimmune-related ILD progressive pulmonary fibrosis; the manuscript is well written but:

I think the authors did not show any novelty in their review, there is already published review and meta-analysis study that addresses the effect of nintedanib in progressive pulmonary fibrosis “Nintedanib in Progressive Pulmonary Fibrosis A Systematic Review and Meta-Analysis

Response: Thank you for your feedback, although there have been several studies investigating the use of Nintedanib for PF-ILDs, our motivation on this study was to narrow down the efficacy of the drug specificaly towards autoimmune related interstitial lung disease, such as those found in Rheumatoid Arhtritis and less commonly SLE in which our study was based on.

Reviewer 2 Report

Comments and Suggestions for Authors

Dear Author ,

Manuscript defines the real-world experience of Nintedanib.

Please respond to the following queries

1. What is progressive pulmonary fibrosis and how will you define in your study

2. The methodology should be clear

3 At what time do antifibrotics should be started in autoimmune disease clarify 

Comments on the Quality of English Language

English is ok 

Author Response

Manuscript titled “THE ROLE OF NINTEDANIB IN REDUCING PROGRESSION OF AUTOIMMUNE DISEASE RELATED INTERSTITIAL LUNG DISEASE”

We would like to express our deep gratitude to the Editor and Reviewers for the important feedbacks and suggestions. We have addressed the Reviewers comments point by point below. We provide clean and tracked changes version of the revised manuscript. Our responses are as following.

Dear Author,

Manuscript defines the real-world experience of Nintedanib.

Please respond to the following queries

  1. What is progressive pulmonary fibrosis and how will you define in your study

Response: Thank you for your feedback, our current understanding of progressive fibrosing ILDs is that it is used as an umbrella term to define diverse group of ILDs with a progressive disease course and clinical features. For the current diagnostic criteria, we defined progressive pulmonary fibrosis as at least two of the three criteria within 12 months: worsening respiratory symptoms; lung function deterioration and increase in fibrotic features in HRCT.  

  1. The methodology should be clear

Response: The methodology to our understanding is clear, as we conducted a mini-review. In the Methods section, we first described that literature searches were performed in PubMed, ScienceDirect, and Scopus using the terms (nintedanib) AND [(ILD) OR (interstitial lung disease)] AND [(progressive) OR (pro-gressivity)] and we applied the PICO framework. All studies found were selected using inclusion and exclusion criteria. Search strategy flowcharts was described in Figure 1.

  1. At what time do antifibrotics should be started in autoimmune disease clarify

Response: Lastly, From the studies and current literature we suggest that prompt treatment of Nintedanib should be started after a progressive pulmonary fibrosis is established.

Reviewer 3 Report

Comments and Suggestions for Authors

In this systematic review, Ardan et al conducted a comprehensive search for relevant randomized-controlled trials, systematic reviews, or meta-analyses on PubMed, ScienceDirect, and Scopus databases. They concluded that nintedanib is a safe option for mitigating the progression of autoimmune disease-related ILD, providing valuable insights into its potential therapeutic role in this context. Overall, the systematic review was conducted in accordance with the procedures, and the conclusions were drawn from the review, which I think are appropriate. No major points are raised. There are a few minor points as follows.

minor concerns)

1) Please provide not only the name of the university or hospital but also the state/province and country of affiliation of the author(s).

2) In Table 3, the authors names were written as "Highland KB, Distler O, Kuwaha M, Allanore Y, Assassi S, et al (2021)" I think "Kuwaha" is a mistake for "Kuwana". Please correct it appropriately.

Author Response

Manuscript titled “THE ROLE OF NINTEDANIB IN REDUCING PROGRESSION OF AUTOIMMUNE DISEASE RELATED INTERSTITIAL LUNG DISEASE”

We would like to express our deep gratitude to the Editor and Reviewers for the important feedbacks and suggestions. We have addressed the Reviewers comments point by point below. We provide clean and tracked changes version of the revised manuscript. Our response are as following.

1) Please provide not only the name of the university or hospital but also the state/province and country of affiliation of the author(s).

Response: Thank you for your feedback, the sugestion that you provided have been implented on our latest edit, we hope that we have not missed anything else

2) In Table 3, the authors names were written as "Highland KB, Distler O, Kuwaha M, Allanore Y, Assassi S, et al (2021)" I think "Kuwaha" is a mistake for "Kuwana". Please correct it appropriately.

Response: Thank you for your feedback, the sugestion that you provided have been implented on our latest edit, we hope that we have not missed anything else

Round 2

Reviewer 1 Report

Comments and Suggestions for Authors

N/A